# Strategies to Improve Resveratrol Systemic and Topical Bioavailability: An Update

**DOI:** 10.3390/antiox8080244

**Published:** 2019-07-25

**Authors:** Sebastiano Intagliata, Maria N. Modica, Ludovica M. Santagati, Lucia Montenegro

**Affiliations:** Department of Drug Sciences, University of Catania, 95125 Catania, Italy

**Keywords:** resveratrol, drug delivery systems, liposomes, lipid nanoparticles, polymeric nanoparticles, cyclodextrins, microemulsions, prodrugs

## Abstract

In recent years, a great deal of attention has been paid to natural compounds due to their many biological effects. Polyphenols are a class of plant derivatives that have been widely investigated for preventing and treating many oxidative stress-related pathological conditions, such as neurodegenerative and cardiovascular diseases, cancer, diabetes mellitus and inflammation. Among these polyphenols, resveratrol (RSV) has attracted considerable interest owing to its high antioxidant and free radical scavenging activities. However, the poor water solubility and rapid metabolism of RSV lead to low bioavailability, thus limiting its clinical efficacy. After discussing the main biochemical mechanisms involved in RSV biological activities, this review will focus on the strategies attempted to improve RSV effectiveness, both for systemic and for topical administration. In particular, technological approaches involving RSV incorporation into different delivery systems such as liposomes, polymeric and lipid nanoparticles, microemulsions and cyclodextrins will be illustrated, highlighting their potential clinical applications. In addition, chemical modifications of this antioxidant aimed at improving its physicochemical properties will be described along with the results of in vitro and in vivo studies.

## 1. Introduction

5-[(*E*)-2-(4-hydroxyphenyl)ethenyl]benzene-1,3-diol (IUPAC name), also known as *trans*-resveratrol (RSV, Figure 1), is a phenolic compound belonging to the stilbenoid group. In particular, the structure of stilbene compounds is characterized by a diarylethene skeleton (C6–C2–C6 unit), generally substituted with hydroxyls (i.e., 3,5,4′-trihydroxystilbene, for resveratrol), methyl, methoxy groups or bound with carbohydrates to provide glycosides [1]. RSV is mainly present in a variety of plants, including grape vine (*Vitis vinifera*), peanuts (*Arachis hypogaea*), Japanese knotweed (*Fallopia japonica*), and foods such as wine, Itadori tea, soy, and red fruits [2,3,4,5,6]. Due to the presence of the central ethylene moiety in its structure, RSV exists as the two possible stereoisomers *cis* and *trans*. Although the naturally occurring RSV is usually the *trans*-isomer (*E*-configuration) [7], under ultraviolet (UV) irradiation *trans*-RSV isomerizes to the *cis*-isomer, which might undergo further photochemical reactions [8,9]. RSV exerts in plants a protective role against UV radiations, injuries, and fungal infections, acting as a phytoalexin [10,11,12,13]. Several studies have investigated the biological activity of RSV by describing a variety of effects on human health [14]. These effects include antioxidant, anticancer, anti-inflammatory activities as well as protection against cardiovascular disease and metabolic syndrome [15,16,17,18].

RSV biological activity has been related to its ability to modulate various biochemical mediators. As reported in the literature, RSV suppresses multiple signaling pathways, including inhibition of phosphodiesterase activity [19,20,21], pro-inflammatory NF-kB-mediated signaling [22,23,24,25], Hedgehog [26,27], and wingless (Wnt) pathways [28]. Conversely, RSV activates various biochemical factors such as adenosine monophosphate (AMP)-activated protein kinase (AMPK) [29,30,31,32,33] and the tumor suppressor p53 [34,35], and also induces nitric oxide synthase (NOS) [36]. Due to its anti-inflammatory action in the adipose tissue [37,38], RSV could be beneficial in the treatment of metabolic diseases [39], obesity and osteoporosis [40,41,42]. Furthermore, it has been shown that RSV could bind alfa and beta estrogen receptors [43,44], and recent studies pointed out RSV’s ability to improve reactive oxygen species (ROS) production in cancer cells acting as a cytotoxic agent [45,46].

In regards to *trans*-RSV physicochemical properties, this molecule is an off-white powder at room temperature (melting point 254 °C) that is mostly soluble in alcohols (solubility decreases with increasing the alcohol carbon chain) [47], and poorly soluble in water (21 μg/mL at pH 7.4 and 37 °C) [48]. Stability and solubility of *trans*-RSV are strongly influenced by pH and temperature [49], and in an aqueous medium, the p*K*_a1-3_ values were found to be 8.8, 9.8, and 11.4, respectively [50]. The overall physicochemical properties of *trans*-RSV (Table 1), as well as the poor solubility in water affect its drug-likeness, including the bioavailability [51,52,53,54]. As reported in the literature, after oral administration, RSV is quickly absorbed at the gastrointestinal level but its bioavailability is lower than 1% due to an extensive first pass effect [53]. Walle et al. reported that RSV plasma half-life was 9.2 h and they observed a first peak of RSV plasma concentration 1 h after an oral dose of 25 mg and a second peak after 6 h [55]. In addition, as a polyphenolic compound, *trans*-RSV undergoes an extensive phase II metabolism, due to the fact that its hydroxyl groups (3,5,4′-positions of the stilbene moiety) are suitable substrates for conjugation reactions (i.e., glucuronidation and sulfation) [56,57]. For all these reasons, many strategies have been attempted to improve both the systemic and topical bioavailability of RSV. Different delivery systems have been proposed as carriers to address the issue of RSV poor water solubility and to improve its stability and permeability through different biological barriers, including lipid and polymeric nanoparticles, liposomes, cyclodextrins, and micro- and nanoemulsions.

On the other hand, medicinal chemists have focused their efforts on two main goals, which include improvement of the pharmacokinetic properties and/or enhancement of the bioefficacy of *trans*-RSV [58,59]. Furthermore, because of the quick accessibility of the *trans*-stilbene scaffold through several synthetic approaches [60,61], as well as the wide range of functionalization that can be performed on phenols, it is no surprise that, so far, a large number of *trans*-RSV derivatives have been developed and reported in the literature [62,63]. Specifically, *trans*-RSV derivatives can be divided into two main groups: (1) prodrugs, (2) synthetic analogs. Each group has been developed for addressing different aims, such as to modulate absorption, distribution, metabolism, and excretion (ADME) of *trans*-RSV (prodrugs), or to enhance its biological and pharmacological activity (*trans*-RSV analogs). From a structural point of view, RSV prodrugs maintain the 3,5,4′-trihydroxystilbene skeleton in their structure and possess suitable promoiety on the phenolic hydroxyls (Figure 2) [64]. Differently, RSV analogs are derivatives obtained by structural modification on the *trans*-stilbene core, for example, modification of the double bond or of the benzene ring, phenyl ring substitutions with halogens or other substituents (Figure 2) [62,65,66,67]. Due to the tremendous amount of publications regarding the *trans*-RSV derivatives and the specific scope of this review, here we mainly focus on the description of RSV prodrugs and technological approaches involving RSV incorporation into different delivery systems.

## 2. Drug Delivery Systems

Since the introduction of the first sustained release formulation in 1952, a great deal of research has been focused on drug delivery systems, leading to the development of three generations of delivery technologies [68]. The main goal of such emerging technologies was to overcome the drawbacks of conventional dosage forms by enhancing drug stability and bioavailability, controlling drug release from the formulation and achieving a drug targeting effect, thus preventing or limiting side effects and improving patient compliance [69].

In the 1950s, the discovery of the existence of microscopic emulsion-like structures in a transparent mixture of oil, alcohol, water and surfactant led to development of the first colloidal delivery systems for pharmaceutical use [70]. Such aqueous systems, containing liquid oil droplets stabilized with different types of surfactants were termed “microemulsions” (Figure 3). However, this term was a misnomer as the droplet size of these systems was in the range 10–150 nm and, therefore, a more appropriate definition would be “nanoemulsions”. Recently, the terms “microemulsions” and nanoemulsion” have been clearly differentiated by McClements, who defined microemulsions as thermodynamically stable systems that form spontaneously, requiring a greater amount of surfactant compared to emulsions, while “nanoemulsions” are kinetically but not thermodynamically stable and they require smaller amounts of surfactants but high-energy input methods for their preparation [71].

In the mid of the 1960s, liposomes, namely vesicles consisting of phospholipid bilayers surrounding one or more aqueous compartments, entered the field of colloidal drug delivery systems due to their many attractive features such as biocompatibility, ability to incorporate both hydrophilic and lipophilic actives, drug targeting and safety [72,73,74,75]. However, some unfavorable properties, including poor physical stability and complex and expensive production methods, have limited the clinical use of liposomes as drug delivery systems.

In the 1970s, polymeric nanoparticles (PN) were developed to obtain drug delivery systems with better stability than liposomes [76,77]. Although PN exhibit several attractive features such as controlled drug release, increased drug solubility, drug targeting, biodegradability, drug protection from degradation and good stability, some disadvantages including toxic degradation and toxic monomers aggregation have limited their practical use as drug delivery systems.

To overcome the drawbacks of liposomes and polymeric nanoparticle while merging their positive features, at the beginning of the 1990s, solid lipid nanoparticles (SLN) were developed [78,79,80]. The great potential of these novel carriers in the development of successful drug delivery systems was immediately realized owing to their many positive features such as ability to incorporate lipophilic and hydrophilic drugs improving their stability and bioavailability, improved drug water solubility, controlled drug release and targeting, safety, biocompatibility, low cost of production and easy scale-up. SLN are aqueous colloidal systems consisting of a solid lipid core that is stabilized by a suitable mixture of surfactants. The SLN preparation method can lead to a different drug distribution in the lipid matrix, generating three types of SLN [81]. In the “solid solution” model, the drug is molecularly dispersed in the lipid matrix, while in the “core-shell” and in the “drug-enriched shell” model, the drug is preferentially incorporated in the core or in the shell, respectively (Figure 3). Unfortunately, due to the crystalline structure of the lipid core, SLN exhibited to main drawbacks: (a) low drug loading, (b) drug leakage during storage. To improve such unfavorable properties of SLN, a second generation of lipid nanoparticles was developed using mixtures of solid and liquid lipids to prevent the formation of a crystalline lipid core. Such lipid nanoparticles, defined as nanostructured lipid carriers (NLC), can show different structures, depending on the type of lipids and the method used for their preparation. As depicted in Figure 3, three types of NLC can be obtained: amorphous type, imperfect type and multiple type [79]. Amorphous type NLC contain a solid amorphous matrix made up of a mixture of solid lipid and specific liquid lipids (medium chain triglycerides or isopropyl myristate). The imperfect type is prepared using a blend of spatially different lipids that lead to the presence of imperfections in the matrix that can host the drug in amorphous clusters. The multiple type NLC contain oil compartments in which the drug can be easily solubilized. Due to the previously mentioned structures, NLC show greater drug loading and better stability in comparison to SLN.

To improve drug solubility, cyclodextrins (CDs), namely cyclic oligosaccharides derived from the degradation of starch catalyzed by the enzyme cyclodextrin glycosyltransferase, have been proposed as drug carriers [82,83,84]. CDs consists of six, seven or eight linked D-glucopyranose units (α, β and γ-CDs, respectively) that are arranged to form a toroidal structure whose external surface is hydrophilic while the internal cavity is lipophilic and it can host lipophilic active ingredients, forming inclusion complexes. Apart from increasing water solubility of lipophilic compounds, drug encapsulation into CDs shows many advantages, as it can result in an increase of drug dissolution rate and bioavailability while decreasing tissue irritation after oral administration, controlled drug release, drug protection from chemical or enzymatic degradation and masking unfavorable drug organoleptic properties (odors or taste). All these advantageous features made drug inclusion into CDs a promising strategy in the field of drug delivery systems.

Owing to its unfavorable physico-chemical properties, easy degradation and poor bioavailability, RSV could benefit from its incorporation into the above-mentioned drug delivery systems. In the following sections, we illustrate the most recent and promising delivery systems that have been investigated as carriers to improve RSV bioavailability after systemic (oral and parenteral) and topical administration.

### 2.1. Systemic Delivery

#### 2.1.1. RSV Oral Delivery Systems

Different types of polymeric and lipid nanoparticles have been used to improve RSV oral bioavailability for the treatment of various pathological conditions, as shown in Table 2. As drug encapsulation into nanoparticles can favor the trans-epithelial drug transport at gastrointestinal level while increasing drug stability and intestinal supersaturated concentration, Singh et al. attempted to improve RSV oral bioavailability using slowly degradable nanoparticles, namely polymeric nanoparticles (NP), which are supposed to be more suitable for oral delivery than rapidly degradable NP (e.g., lipid nanoparticles) and non-degradable NP (e.g., inorganic NP) [85]. These authors prepared RSV loaded poly (dl-lactide-*co*-glycolide) (PLGA) NP and evaluated RSV pharmacokinetics, in vivo bio-distribution and in situ single-pass intestinal perfusion from an optimized formulation after oral administration in rats. The results of this study pointed out that delivering RSV via this nano-particulate system allowed for overcoming the enterohepatic recirculation and improved RSV oral bioavailability in comparison to the free drug. Owing to polymeric NP poor bio-adhesion and the rapid mucus turnover at gastrointestinal level, drug absorption from these nano-carriers could be limited. Therefore, Siu et al. designed NP consisting of poly(lactic-*co*-glycolic acid) (PLGA) on which they performed a galactosylation process by introducing *N*-oleoyl-d-galactosamine to obtain galactosylated PLGA NP (GNPs) that were expected to show better bio-adhesion and, therefore, better drug absorption at gastrointestinal level than conventional NP [86]. Loading RSV in such GNPs, the authors observed an increase of RSV oral bioavailability up to 335.7% in comparison to RSV suspensions after administration in rats, which was attributed to an increase of the intestinal permeability and transcellular transport of RSV. In addition, RSV GNPs showed a better anti-inflammatory activity in RAW 264.7 cells model than free RSV and RSV loaded into non-galactosylated PLGA NP [86].

As chitosan is a natural, non-toxic and highly biocompatible polymer, several researchers prepared NP based on chitosan derivatives to improve RSV oral delivery. Carboxymethyl chitosan (CMCS) was used to prepare RSV loaded NP that were evaluated for their ability to improve RSV physicochemical properties, antioxidant activity and in vivo bioavailability after oral administration in rats [87]. Such an NP proved to be a promising carrier for RSV as they increased RSV water solubility and antioxidant activity, showing a 3.5-fold increase of in vivo relative bioavailability in comparison to the free drug [87].

In another study, *N*-trimethyl chitosan conjugated with palmitic acid (TMC-g-PA) was used as surface modifier of RSV loaded SLN to improve the oral delivery of this drug [88]. After oral administration in Balb/c mice, these nanoparticles provided a 3.8-fold increase of RSV bioavailability compared to the free drug. Such an increase was attributed to different factors such as muco-adhesive and high absorption properties of the polymer, and an ability to prevent RSV enzymatic and/or chemical degradation due to its loading in this type of NP [88].

Although lipid nanoparticles are rapidly degraded and they cannot be regarded as the best choice for oral drug delivery, some researchers investigated the feasibility of using SLN and NLC as nanocarriers for RSV oral administration. Pandita et al. designed SLN made up of stearic acid and coated with poloxamer 188 as carriers to improve the oral bioavailability of RSV [89]. These RSV loaded SLN showed good technological properties and led to an enhancement of RSV bioavailability in comparison to the free drug after oral administration in rats. The authors concluded that the use of these colloidal nanocarriers could be regarded as promising strategy to improve RSV effectiveness after oral dosing [89]. Similar conclusions have been reported by Neves et al. who prepared RSV loaded SLN and NLC and, after characterizing these nanocarriers, performed an in vitro simulation of gastrointestinal transit [90]. The results of such experiments showed that RSV remained mostly associated to both type of lipid nanoparticles after their incubation in digestive fluids, leading the authors to suggest that SLN and NLC could be promising RSV oral delivery systems due to their ability to prevent RSV degradation and to provide a controlled drug release after their administration [90].

Recently, a novel approach has been explored by Santos et al. who prepared layer-by-layer (LbL) self-assembly NP loading RSV and performed a pharmacokinetic study in rat to assess the usefulness of such carriers as RSV oral delivery systems [91]. This study provided very encouraging results, showing a significant increase of RSV chemical stability and oral bioavailability.

#### 2.1.2. RSV Parenteral Delivery Systems

As previously mentioned, RSV has been proposed as anti-cancer agent due to its many biological properties. However, RSV therapeutic use in the treatment of different types of tumors is strongly impaired by its unfavorable physicochemical properties such as poor water solubility and easy chemical and enzymatic degradation that prevent its parenteral administration [92]. To overcome such drawbacks, several researchers have attempted to improve RSV effectiveness as anti-cancer agent loading this molecule into specifically designed delivery systems.

Song et al. co-encapsulated RSV and docetaxel (DTX) into lipid-polymer hybrid NP (LPNs) for the treatment of lung cancer [93]. LPNs were designed for targeted delivery of RSV and DTX to the mitochondria of tumor cells by conjugating the epidermal growth factor (EGF) and stearic acid (SA) with polyethylene glycol (PEG). The resulting polymer (EGF-PEG-SA) was used to prepare EGF LPNs using a nanoprecipitation technique. In vitro assays highlighted the greater effectiveness of RSV/DXT loaded EGF LPNs in decreasing the viability of tumor cells (HCC827 and NCIH2135) in comparison to RSV/DXT loaded LPNs, thus confirming the usefulness of including the targeting moiety EGF in the NP. In addition, in vivo studies performed administering RSV/DTX EGF LPNs intravenously in mice bearing lung cancer showed a significant inhibition of tumor growth and a notable reduction of its size [93].

PEG and polylactic acid (PLA) were used to prepare polymeric nanoparticles to improve RSV stability and to achieve a controlled release for in vivo cancer treatment [94]. In vitro assays carried out on CT26 colon cancer cells pointed out a significant reduction of cell number and colony forming capacity due to the treatment with RSV-NP in comparison to controls. After intravenous administration of RSV PEG-PLA NP in CT26 tumor-bearing mice, a significant decrease of tumor growth with a concurrent increase of the survival time of mice was observed, suggesting that incorporating RSV into NPs could be a useful strategy to improve RSV effectiveness in cancer therapy [94].

Guo et al. modified PEG-PLA NP including in these NP transferrin (Tf), whose receptor is expressed in brain capillaries, in order to increase NP ability to target glioma cells [95]. RSV loaded Tf- PEG-PLA NP exhibited an increased anti-cancer activity after intraperitoneal administration in C6 glioma-bearing rat models, leading to a reduction of tumor volume with a concomitant increase of survival time [95]. Figueirό et al. addressed the issue of improving the RSV antiglioma effect by loading RSV into lipid core nanocapsules (LCN) made up of poly(ε-caprolactone), capric/caprylic triglyceride and sorbitan monostearate [96]. Rats bearing brain-implanted C6 gliomas, who were intraperitoneally administered with RSV loaded LCN, showed a remarkable amelioration of different parameters related to the tumor expression (intratumoral hemorrhaging, peritumoral edema, peripheric pseudopalisading), including a notable decrease of tumor size. The authors attributed these results to a better RSV transportation across the blood brain barrier and a lower RSV binding to plasma protein when loaded into LCN [96].

Several authors have proposed the use of polymeric micelles (PM) as alternative to polymeric nanoparticles for RSV deliver in the cancer therapy, due to several advantages of PM such as pharmacokinetic improvement of the loaded drug and ability to deliver their payload at the target site, thus reducing systemic side effects [97]. In some studies involving the use of polymeric micelles as delivery system, RSV was co-loaded with DTX, paclitaxel (PTX) or heme oxygenase-1 gene (pHO-1) [98,99,100]. PM co-loading RSV and DTX were obtained using a specifically designed and synthesized polymer, namely methoxyl poly(ethylene glycol)-poly(d,l-lactide) copolymer (mPEG-PDLA) and were assayed to evaluate their cytotoxicity in MCF-7 cells and their pharmacokinetic parameters after intravenous administration in rats. In this study, a strong synergic effect between RSV and DTX co-loaded into PM was observed along with an increase of area under the curve (AUC) values of these drugs [98]. An analogous synergic effect between RSV and PTX was reported by Hu et al. on PTX-resistant human lung adenocarcinoma epithelial (A549/T) cell line and mice sarcoma 180 (S180) cells [99]. After an 8-day treatment by intravenous injection of RSV and PTX co-loaded PM in S180 solid tumor bearing mice, a greater inhibition of tumor growth was observed in comparison to PM loading only a single drug (RSV or PTX). Kim et al. prepared self-assembled polymeric micelles to simultaneously deliver RSV and pHO-1 [100]. Cholesterol-conjugated polyamidoamine (PAM-Chol) was synthesized and used to prepare self-assembled micelles loading both RSV and pHO-1. Such micelles significantly reduced the level of pro-inflammatory cytokines after addition to lipopolysaccharide (LPS)-activated macrophage cells, thus showing a notable anti-inflammatory activity. The administration of RSV and pHO-1 loaded PAM-Chol micelles by inhalation in Balb/c mice resulted in the inhibition of the nuclear translocation of NF-kB and in the decrease of pro- inflammatory cytokines in lungs, thus highlighting the feasibility of using the above-mentioned formulations as carriers to deliver RSV and pHO-1 at the lung level (see Table 3).

Karthikeyan et al. proposed the use of gelatin NP to improve the therapeutic effectiveness of RSV [101]. After intravenous administration of RSV-loaded gelatin NP in mice, RSV serum levels about two-fold higher than that obtained with free RSV were observed along with a longer RSV plasma half-life. Besides the improvement of RSV bioavailability due to its loading into gelatin NP, the authors pointed out a greater anti-proliferative efficacy of these RSV loaded NP in NCI-H460 lung cancer cells in comparison to free RSV [101].

Chitosan-coated lipid nanoparticles have been investigated as carriers to improve RSV ability to cross the blood brain barrier via nasal delivery [102]. In this study, an increase of RSV concentration in rat cerebrospinal fluid (CSF) in comparison to free RSV was observed, thus suggesting that this type of carrier could be successfully used for drug delivery to the brain by nasal administration.

While RSV loading into different types of nanoparticles or micelles provided encouraging results as regards its potential therapeutic applications, the attempt to use cyclodextrins as delivery system for RSV proved unsuccessful. To improve RSV water solubility, Das et al. prepared RSV inclusion complexes with hydroxypropyl-β-CDs (HP-β-CDs) and randomly methylated-β-CDs (RM-β-CDs), which were intravenously and orally administered, respectively, in rats [103]. The authors observed an improvement of RSV oral absorption rate that was attributed to an increase of RSV water solubility but no RSV bioavailability improvement was obtained both after oral and intravenous administration of RSV complexes with CDs [103]. Parental RSV drug delivery systems are summarized in Table 3.

### 2.2. Topical Delivery

To date, two types of colloidal delivery systems have been mainly investigated to improve RSV topical delivery, namely lipid nanoparticles and liposomes (Table 4).

As RSV *trans*-*cis* isomerization is regarded as a limiting factor in its topical use, to improve RSV stability, Carlotti et al. designed SLN containing tetradecyl-γ-cyclodextrin and evaluated RSV in vitro penetration through pig skin from these nanocarriers [104]. Such a delivery system improved RSV photostability, providing a notable RSV accumulation in the skin and increased anti-lipoperoxidative activity [104].

Chen et al. prepared RSV loaded NLC using cetyl palmitate as solid lipid and sesame oil as liquid lipid to favor RSV skin permeation [105]. After performing in vitro percutaneous absorption experiments on human skin, these authors pointed out an increase of RSV skin permeation from SLN, suggesting that these nanocarriers could be successfully used to improve RSV topical effectiveness [105].

A recent work evaluated the effectiveness in the treatment of irritant contact dermatitis (ICD) of RSV loaded SLN in gel vehicles [106]. The authors performed both in vitro skin permeation experiments on human skin and in vivo studies on an ICD-induced BALB/c mice model to assess the ability of such formulations to improve RSV skin penetration and to inhibit ear swelling, respectively. The results of this investigation revealed that RSV loading into SLN provided about a three-fold increase of its retention in the skin layers (epidermis and dermis) in comparison to free RSV and a gel formulation containing RSV loaded SLN was as effective as a marketed formulation based on corticosteroids in reducing tissue oedema. Therefore, the authors concluded that RSV loaded SLN could be a suitable therapeutic alternative in the treatment of ICD [106].

RSV loaded SLN with and without soy phosphatidylcholine (SPC) were tested to evaluate their effect on RSV in vitro permeation through pig skin and their ability to inhibit tyrosinase activity in vitro [107]. SLN without SPC provided greater skin permeation of RSV and higher inhibition of tyrosinase activity than SLN containing SPC. As RSV loaded SLN showed a higher percentage of tyrosinase inhibitory activity than kojic acid, a well-known tyrosinase inhibitor, the authors concluded that this colloidal formulation could be useful in the topical treatment of several skin disorders [107].

A further study, performed in vivo on healthy volunteers, highlighted the ability of RSV loaded SLN to provide a greater skin hydration in comparison to RSV loaded NLC [108]. These findings were attributed to the higher crystallinity degree of SLN in comparison to NLC. A comparison between SLN and NLC ability to favor RSV targeting into the different skin layers has been performed by Gokce et al. carrying out in vitro penetration studies on rat skin [109]. In this work, it was highlighted that SLN provided a greater RSV accumulation in the epidermis, while NLC led to a greater amount of RSV in the dermis. The authors speculated that the smaller size of NLC in comparison to SLN was a key factor in determining a better penetration of RSV in the deeper skin layers [109].

In contrast, the work by Park et al. on chitosan-coated liposomes as topical delivery system for RSV did not point out a significant influence of the nano-carrier size on RSV skin permeation [110]. As illustrated in this work, although chitosan coating increased liposome mean sizes in comparison to uncoated liposomes, chitosan-coated liposomes provided a greater RSV in vitro permeation through mouse skin, likely due to their positive charge that facilitated their interaction with the negatively charged skin surface [110]. Chitosan-coated liposomes have also been proposed as RSV delivery systems for the topical treatment of vaginal inflammations and infections due to the ability of such RSV liposomal formulations to provide stronger in vitro antioxidant and anti-inflammatory activities than RSV solutions [111,112].

To address the issue of RSV poor topical bioavailability, different research groups designed ultra-deformable liposomes (UDL), namely vesicles made up of phospholipids and edge activators that destabilize the lipid bilayer increasing its deformability. Doppalapudi et al. prepared psoralen and RSV co-loaded UDL, whose free radical scavenging activity was determined by in vitro anti-oxidant assays, while their potential in the topical treatment of vitiligo was assessed using B16F10 cell line. Such co-loaded UDL stimulated tyrosinase activity to a greater extent than the control while retaining the antioxidant activity [113].

Cosco et al. designed UDL as carriers for topical delivery of RSV and 5-fluororacil (5-FU) [114]. In vitro permeation of RSV and 5-FU from UDL was assess through excised human skin while the anti-cancer activity of such co-loaded UDL was evaluated on SK-MEL-28 and Colo-38 cells. The results of this study highlighted the good potential of RSV and 5-FU co-loaded UDL in the treatment of non-melanoma skin cancers [114].

## 3. Prodrugs

Prodrug approach is a successful medicinal chemistry strategy and a popular method in the modern drug discovery that aims to improve the physicochemical properties or modulate the pharmacokinetic parameters of a drug [115,116]. This strategy has become a useful and powerful tool to overcome formulation and delivery issues during the drug development process [117]. Frequently, these problems are correlated with poor aqueous solubility, or more often, with the inability to pass the biological barriers, which drastically affect the bioavailability and bioefficacy of a bioactive agent [117,118]. In addition, as in the case of RSV, chemical instability or extensive metabolism can also limit the activity in vivo [54,119,120].

As mentioned above, the phenolic hydroxyls rapidly undergo conjugation reactions; indeed, the conversion in different chemical entities (e.g., esters, sulfonates, phosphates, carbamates) represents an effective way to mask and protect them from metabolism [64]. Moreover, the transformation of a labile chemical group into a more stable one can also result in a better distribution, permeability, and absorption, making the prodrug a more suitable molecule for different administration routes in comparison to the parent compound. However, it is worth to note that the clinical translation of a prodrug is a more challenging and complicated process than that of standard drugs, due to the many variables involved during its development, such as synthetic accessibility, analytical profiling, site and rate of bioconversion, metabolism, and toxicity profile of the prodrug and the possible by-products [116].

### 3.1. Prodrugs for Systemic Delivery of RSV

#### 3.1.1. Prodrugs to Improve the Bioavailability of RSV

Many RSV prodrugs have been only assayed for their technological properties or for their ability to improve the bioavailability of RSV (Table 5); thus, their therapeutic potential was disregarded. Biasutto et al. at the University of Padova, reported several RSV derivatives carrying different capping groups at the phenolic hydroxyls [59,64]. In 2009, a series of triesters and tri-methanesulfonyl derivatives of RSV (Figure 4), such as RSV triacetate (**1**), RSV tri-mPEG1900 (**2**), and RVS trimesylate (**3**), respectively, were synthesized and evaluated for their in vitro stability [121]. The ester derivatives, compounds **1** and **2**, were hydrolyzed both in aqueous media and in blood, while the RVS trimesylate derivative (**3**) was extremely stable in any condition. Similar results were obtained when the epithelial absorption of the title compounds was tested in ex vivo permeation experiments on rat intestine. A mixture of phase II metabolites of RVS triacetate (glucuronosyl- and sulfate- derivatives) was found in the basolateral compartment, suggesting inefficient protection of the carboxy ester moieties. On the other hand, when the PEGylated derivative was provided on the apical side, a non-negligible amount of not conjugated RSV (about 50%) reached the basolateral side; however, the same result was obtained using a mixture of RSV and PEG (most likely an adjuvant effect). Finally, RVS trimesylate passed from the apical to the basolateral side without modification [121].

In contrast to this study, Liang et al. reported that the pharmacokinetic properties of RSV triacetate in vivo (rats) were improved compared with RSV [122]. In particular, following a single intragastric gavage (i.g.) administration, the half-life (t_1/2_) and the plasma concentration curve (AUC) were higher than RSV with a significant distribution observed in liver, spleen, heart, and lung [122].

Esterification of RSV with α-d-glucose, through a succinyl linker, gave 3,5,4′-tri(α-d-glucose-3-*O*-succinyl)-resveratrol (**4**, Figure 4), a derivative of RSV with improved solubility in water [123]. The intragastric administration of **4** in rats showed a maximum concentration of the major metabolites (RSV sulfate, and RSV glucuronide) after four hours, and no relevant differences in plasma AUC if compared with RSV administration. The newly synthesized prodrug was quite stable in the gastric and intestinal-like conditions, even though it was quickly metabolized in blood [123].

A succinyl linker was also used to develop different PEG-resveratrol prodrugs, bearing the MeO-PEG succinyl ester or the MeO-PEG succinyl amide moieties in the 4′- or 3-position of the RSV scaffold (**5a**–**f**, Figure 5) [124]. In this study, the authors described synthesis and development of an analytical method useful to monitor and characterize the process of PEG conjugation to RSV, however, no investigation of the new prodrugs neither in vitro nor in vivo was performed [124].

Further PEG-resveratrol conjugates, containing succinyl or acetyl group and various amino acids as linkers, were designed and synthesized with the aim to improve the solubility and bioavailability of RSV (**6a**–**l**, and **7a**–**l**, Figure 5) [125]. Preliminary in vitro characterization showed that the solubility in water of the new set of compounds was higher than that of RSV (>900 mg/mL vs. 0.03 mg/mL), with a satisfactory controlled release observed for all of them (52.2–99.1% and 78.5–29.6% over a 72 h period, with or without the presence of lipase, respectively). The authors suggested that the release performance might be influenced by the steric hindrance of the group of each amino acid and, therefore, by the intrinsic stability of the ester bond [125]. Consequently, in this work, conjugation of PEG with amino acids seemed to be an effective strategy to improve both the solubility and bioavailability of RVS; however, the tests were only performed in vitro, and further in vivo pharmacokinetic studies are needed to confirm its validity.

Larrossa et al. reported a series of RSV glucoside derivatives, inspired by the natural stilbenoid *trans*-piceid [126]. The new RSV prodrugs brought either one or two glycosyl groups at the phenolic hydroxyls (specifically at 4′-, 3,5-, or 3,4′-position). Moreover, the original *trans*-piceid structure (3-*O*-glucoside) was further modified by adding fatty acid groups of different length at 6-position of the sugar. Subsequently, compounds **8**–**10** (Figure 6) were selected for both in vitro and in vivo studies. In order to evaluate their chemical stability, the new derivatives were incubated for six hours with human colon cancer Caco-2 cells. As a result, no conjugation derivatives from phase II metabolism were found, suggesting an improvement in metabolic stability for the new RSV glucoside derivatives. In particular, the most stable was the 3,5-di-*O*-glucosyl, compound **8**, which showed the highest rate of the remaining compound after twenty-four hours of incubation. Similarly, the acyl modification at 6′-position of the glucosyl moiety slowed down the metabolism for both compounds **9** and **10**. In vivo metabolic studies, in healthy mice, outlined that the maximum concentration of RSV in the colon was reached after four hours from its oral administration, while higher doses of RSV were found after oral administration of compounds **9** and **10**, suggesting that glucoside derivatives might deliver RSV effectively to the colon [126]. The same research group, very recently, described new alkylated resveratrol derivatives, and alkylated resveratrol prodrugs as potential therapeutics for neurodegenerative diseases [127].

A series of RSV prodrugs, in which the acetal groups were used to link the phenolic hydroxyls with short ethylene glycol oligomers (OEG), was described in 2013 (Figure 7) [128]. In this set, compounds with four or six ethylene glycol (EG) units showed better solubility in water than the compounds containing three units, with an improvement of the pharmacokinetic properties of RSV. However, adsorption was also affected by the nature of acetals. Specifically, compounds with two hydrogens as substituent were too stable (**11a-g**), as only the prodrugs were detected in the blood. On the other hand, the compound with two methyls as substituent (**12**) gave opposite results. Finally, the compound with one hydrogen and one methyl (**13**) gave good results in term of protracted regeneration of the parent molecule after its absorption, although this result was lower than that of RSV [128].

Several carbamate derivatives of RSV with high water solubility were developed and reported [129,130,131,132,133]. *N*,*N*-disubstituted derivatives, bearing as a substituent on each nitrogen atom a methyl group and either a butyl-glucosyl or a methoxy-polyethylene glycol 350 (mPEG 350) moiety were described in 2014 (**14** and **15**, Figure 8) [129]. Both analogs showed greater solubility in water than RSV (>50 mM), and in ex vivo absorption studies on rat intestine, compound **14** was able to cross the intestinal wall efficiently without undergoing metabolic transformations, while neither compound **15** nor its metabolites were detected in the basolateral side. Due to the high hydrophilicity of compound **14**, which would not allow passing the cell membranes through passive diffusion, the authors suggested the possible involvement of an active transport mechanism mediated by glucose transporters. Subsequently, in vivo pharmacokinetic studies were performed, and no detectable amount of compound **14** was found in the bloodstream, suggesting that the *N*,*N*-disubstituted derivatives of RSV were too stable to be used as prodrugs [129].

In a second attempt, the hydrolytic stability of carbamate derivatives of RSV was modulated by varying the substitution pattern of the nitrogen atom (e.g., *N*-monosubstituted carbamates rather than *N*,*N*-disubstituted ones). Azzolini et al. designed and synthesized several *N*-monosubstituted carbamoyl-RSV derivatives (Figure 9), in which all or part of the phenolic hydroxyls of RSV were connected via carbamoyl linker to dihydroxypropyl (DHP) and 6-deoxygalactosyl (DGAL) moieties [130]. As a result, the introduction of both promoieties (glycerol or galactose) led to water-soluble compounds, which were stable in acid solution (gastric pH). Moreover, the rate of hydrolysis was conveniently slow at intestinal pH and in blood. Surprisingly, the tri-DGAL substituted derivative, compound **16** (Figure 9), was not absorbed in the gastrointestinal tract. However, the authors found that chronic administration of this derivative resulted in the presence of its partially hydrolyzed derivatives in the colon, which could represent an alternative for the use of glycosyl-RSV derivatives for the treatment of colitis or to prevent colon cancer [130].

The same research group described another series of *N*-monosubstituted carbamoyl-RSV derivatives bearing methoxy-oligo(ethylene glycol) groups of different chain length (*n* = 3, 4, and 6) as promoieties of the new prodrugs (Figure 10) [131]. The use of short EG chains was a strategy to balance both the overall physical-chemical properties and the loading capacity of the new derivatives. The new carbamate derivatives **17**–**19** showed remarkable stability under acid conditions. Following intragastric administration to rats, the prodrugs were detected in the bloodstream, intact or partially metabolized, thus confirming a specific resistance against the first-pass metabolism. The length of the EG chain influenced the absorption, in particular, compounds **18** and **19** (with four and six EG units, respectively) had a better absorption than compound **17** (three EG units) [131].

Similar results were obtained with amino acid carbamates [132]. This new set of RSV derivatives possessed a natural amino acid moiety (i.e., Leu, Ile, Phe, and Thr, respectively) attached by carbamoyl linker (**20**–**23**, Figure 11). In this work, the new prodrugs showed, once again, good resistance to hydrolysis in aqueous media and blood, confirming that carbamates are suitable as a capping groups to obtain RSV prodrugs. However, in this specific case, the presence of an extremely hydrophilic promoiety, such as an amino acid, led to insufficient absorption of the derivatives. The authors suggested that one of the reason for the failure in passing the inner membrane might be the large and ramified structure of the molecules that did not allow them to be carried by transporters [132]. To overcome this issue, another series of amino acid carbamates, in which the 3,5-dimethoxy-RSV (pterostilbene) was used as a starting material, has been synthesized in 2017 [133]. Specifically, the metabolic soft spot of pterostilbene (the 4′-position) was masked by a carbamoyl bond-based amino acid group (**24a**–**i**, Figure 11). The new set of RSV derivatives were readily absorbed after intragastric administration in rats, and among them, compound **24a** (incorporating isoleucine) showed an optimal pharmacokinetic profile. Further characterization of **24a** showed that after several hours from its administration, the absorption and concentration of pterostilbene were higher than that of pterostilbene administrated as such. Moreover, further studies performed on an in vitro model of human intestinal absorption revealed that, unlike **17–19**, compound **24a** could cross the apical membranes either through passive diffusion or by an active transport mechanism [133].

#### 3.1.2. Prodrugs to Improve the Pharmacological Activity of RSV

Aside from their ability to improve the bioavailability of RSV, several RSV prodrugs have been developed and tested for their pharmacological properties, especially for anticancer and antioxidant effects (Table 6).

New conjugated derivatives of RSV or RSV diacetate with a triphenylphosphonium group (TPP) were synthesized (**25** and **26**, Figure 12), to increase the pro-oxidant action of RSV on mitochondria [134]. The presence of a membrane-permeable lipophilic cation, like TPP, drove the molecule to the mitochondrial matrix, which had a negative potential. Among them, **26** showed a fast-growing cytotoxicity effect on murine colon cancer cell line [134]. Subsequently, Sassi et al. synthesized and characterized different analogs, bearing the TPP group at 4′- and 3-position (**27**–**30**, Figure 12) [135]. The cytotoxicity of the new derivatives was higher for the methylated ones, and the cell death was mainly of the necrotic type and was due to the H_2_O_2_ production upon the accumulation of the compounds into mitochondria. It is worth noting that RSV at the same dose range (low μM) did not show any effect, confirming that the accumulation in mitochondria was due to the presence of TTP group [135]. Moreover, further studies on the mechanism of generation of reactive oxygen species (ROS) and mitochondrial depolarization suggested an interaction with the respiratory chain complexes, which might cause superoxide production by these agents. The authors also speculated about the possibility for these derivatives to involve the ATP synthase into a conduit for uncoupled proton translocation [136].

The combination of two different active moieties within the same molecule represents an attractive strategy to exert a synergic effect and avoid drug-drug interactions [137,138,139,140]. Bernhaus et al. synthesized the 3,5-*O*-digalloylresveratrol (DIG) **31** [141], a diester of RVS with the gallic acid (Figure 13), a natural product compound showing pro-apoptotic properties against various tumor cell lines [142,143,144]. In this study, the biological effects of **31** in HT-29 human colon cancer cells were observed. Treatment of the cancer cells line for 72 h with 40 µM of **31** produced a dose-dependent apoptotic effect (45% of cells). Interestingly, **31** inhibited the growth of HT-29 cells more effectively than the combination of RSV and gallic acid in a ratio of 1:2. Moreover, the anticancer properties of **31** were supported by its inhibitory effect observed on ribonucleotide reductase. Specifically, **31** produced arrest of the G0–G1 cell cycle and depletion of S and G2-M phase cells in HT-29 tumor cells. Finally, simultaneous treatment with **31** and 5-FU (an established drug for colon cancer), enhanced the growth inhibitory effects in HT-29 cells. Unfortunately, the in vitro stability studies performed on **31** showed its rapid hydrolysis into RSV [141].

Zhu et al. studied the antiproliferative properties of a series of RSV-based aspirin prodrugs [145]. Intending to release RSV and aspirin concurrently, Zhu and colleagues synthesized the RSV-aspirin hybrid (RAH) **32**, by esterification of the 4′-phenolic hydroxyl of RSV with aspirin (acetylsalicylic acid) (Figure 13). The regioisomer of **32**, resulting from esterification of the 3-phenolic hydroxyl of RSV, was also obtained with a ratio of 2.3:1. The anticancer properties of these two compounds were evaluated in HCT-116 and HT-29 human colon cancer cells using the MTT assay. As a result, both compounds inhibited the growth of cancer cells in a dose-dependent manner. The observed effect was higher than that showed by the administration of RSV and acetylsalicylic acid individually or simultaneously (equimolecular amount). In particular, **32** was more effective than its regioisomer, suggesting that the position of esterification influenced the anticancer activity.

Moreover, **32** decreased the levels of cyclins D1 and E, which interrupt the cellular cycle, in a dose-dependent manner over twenty-four hours. Besides, **32** exerted a pro-apoptotic effect, due to the activation of the caspase-3 in the mitochondrial pathway. Metabolic in vitro and in vivo studies on **32** were also performed. After incubation of **32** in HCT-116 cancer cell line, a deacetylation reaction occurred, and the maximum concentration of the metabolite of **32** was reached after four hours. Subsequently, over the twenty-four hours, further hydrolysis of deacetyl-RAH into RSV and salicylic acid was observed. In vivo studies performed in mice, using LC-ESI mass spectrometry, revealed that the majority of **32** crossed the stomach intact, and then RVS, deacetyl-RAH, and acetylsalicylic acid were released in the intestine or the plasma after absorption through intestinal enterocyte, while a small amount of **32** reached the target cells [145].

Other RSV-salicylate hybrid derivatives were developed to inhibit the CYP1A1, which is responsible for the activation of environmental procarcinogens [146]. In this study, Aldawsari et al. observed a significant inhibition of the CYP1A1 activity and a decrease in CYP1A1 mRNA exerted by the new RSV-salicylate hybrids. Interestingly, under the same experimental conditions, RSV did not show inhibition properties of the cytochrome, supporting the development of hybrid molecules as chemopreventive agents [146]. In another work, they also evaluated the same compounds against the DNA-methyltransferase inhibition, thus showing a cytotoxic effect in HT-29, HepG2, and SK-BR-3 human cancer cells higher than that of RSV [147].

A newly synthesized carbonate derivative of RSV (**33**, Figure 13) was reported by Goldhahn et al. who compared the anticancer effects of RSV and **33** against Jurkat leukemia CD4^+^ T-cells [148]. In this study, compound **33** showed an improvement of both the antiproliferative and pro-apoptotic activity in the selected leukemia cell line compared to RSV, even though, unlike the RSV the apoptotic effect of **33** was not mediated by the caspase-3/PARP signaling cascade. Moreover, **33** was 3-fold more potent than RSV in blocking IL2 gene expression. Despite these results, the authors did not elucidate the signaling pathway involved in the pro-apoptotic activity of **33** [148].

The synthesis of three 4′-RSV ester derivatives (**34**–**36**, Figure 14) along with their in vivo antidepressant activity evaluation was reported [149]. Initially, the behavioral profile of the tested compounds was performed to establish their safety and toxicity characteristics, using Long-Evans rats. Subsequently, the standard Porsalt forced-swim test was used to assess the anti-depressant activity of **34**. Treatment with either 20 mg/kg, and 90 mg/kg (s.c.) of 4′-acetyl-RSV showed a significant decrease in depressive-like behaviors of rats if compared to the control, however neither the 4′-benzoyl-RSV (**35**) nor the 4′-pivaloyl-RSV (**36**) were evaluated [149].

More recently, Peterson et al. studied the effects of 4′-RSV ester derivatives (**36**–**38**, Figure 14) on calcium dynamics in triple-negative breast cancer (TNBC) [150], a very aggressive subtype of breast cancer so called because the three common molecular markers (estrogen receptors, progesterone receptors and HER2) are missing. The rational design for this study started from the consideration that RSV could stimulate apoptosis in cancer cell through an increase of Ca^2+^ level, and that during TNBC specific changes in the Ca^2+^ signaling increase the growth of the tumoral cells. Compounds **36**–**38** were obtained by esterification of the 4′-hydroxy function of RSV with the pivalate, isobutyrate, or butyrate group, respectively (Figure 14). These compounds were tested for their ability to decrease cell viability in MDA-MB-231 cancer cell lines, using an MTT assay. As a result, both compounds, the 4′-pivalate RSV (**36**) and 4′-isobutyrate RSV (**37**) were more effective than RSV in reducing cell viability (14.14% and 7.70% vs. 58.45%, respectively). However, this study revealed that the effects exerted by compounds **36** and **37** were not mediated by the Ca^2+^ signaling pathway, like in the case of RSV [150].

The in vitro antioxidant properties for a new series of RSV prodrugs was reported by Oh and Shahidi [151]. The new RSV ester derivatives **39a**–**l** (Figure 14) were obtained by esterification of RSV with acyl chlorides of different chain length (propionyl, butyryl, caproyl, capryloyl, capryl, lauroyl, myristoyl, palmitoyl, stearoyl, oleoyl, eicosapentaenoyl, and docosahexaenoyl chloride, respectively). Different acyl chlorides were selected due to the beneficial effect for human health of short-chain, and medium-chain saturated fatty acids. For examples, antitumor and cardioprotective properties are attributed to eicosapentaenoic and docosahexaenoic acids; propionic and butyric acids are endowed with antitumor properties on colon and ability to inhibit cholesterol production, whereas caprylic, capric, and lauric acids are useful as an energy source. Therefore, esterification of RSV with various fatty acid moieties could result in synergic effect or could modify the pharmacokinetic properties of the parent compound. HPLC-MS studies were performed to establish the structure of these ester prodrugs, which have been obtained as mixtures of 4′- or 3- monoester (primary products), and 3,4′- or 3,5-diesters. The antioxidant properties of these derivatives were evaluated through DPPH radical and ABTS radical cation scavenging tests. All derivatives in both tests demonstrated a lower scavenging effect than RSV, which could be due to the absence of the hydroxyl functions required for the antioxidant effect [151]. In another paper, the same authors reported that compounds **39a**–**l** exhibited hydrogen peroxide scavenging activity higher than RSV [152]. Moreover, the tested compounds were able to inhibit DNA scission induced by hydroxyl radical. All derivatives, except RSV ester with eicosapentaenoic acid, showed inhibition properties on LDL oxidation induced by copper [152].

### 3.2. Prodrugs for Dermal Delivery of RSV

The topical usage of RSV in pharmaceutics and cosmetics has been evaluated, and a large number of in vitro and in vivo studies regarding the effects of RSV on the skin have been reported [153,154]. Even though phase II biotransformations (glucuronidation, sulfation, etc.) also occur in cutaneous tissues [155], the whole-skin intrinsic metabolic clearance is lower than the liver one; thus the dermal usage of RSV might result more effective than its systemic administration. However, as skin’s metabolism remains relevant, the use of RSV prodrugs for skin diseases or for aesthetic indications (e.g., anti-aging and skincare) could have some advantages. For these reasons, the interest in using RSV prodrugs for dermal applications has been growing significantly (Table 7) [156,157,158,159].

In 2007, the stability of topically applied RSV was improved by developing RSV triphosphate (**40**, Figure 15), thus reducing its susceptibility to degradation and releasing the parent compound by epidermis enzyme phosphatases [160]. Analysis of Raman spectra showed dephosphorylation of **40** in the stratum corneum; however, the conversion to RSV did not occur when skin samples were exposed to steam, due to the heat labile nature of enzymes. Moreover, the spatial distribution and time-dependence permeation of both RSV and **40** were evaluated. As a result, the higher water content in the viable epidermis allowed a better permeation of **40**, which is more soluble in water than RSV. Moreover, since the dephosphorylation process was also observed in the viable epidermis, application of RSV as triphosphate prodrugs resulted in a more homogenous distribution of the active compound throughout the stratum corneum and the viable epidermis [160].

The synthesis and whitening effect of RSV triacetate (**1**, Figure 4) was described [161]. Initially, compound **1** was tested for its chemical stability (PEG solution stored at 60 °C), and the anti-melanogenic effect in vitro. The tri-acetylated derivative was more stable to oxidative discoloration than the parent compound. The regeneration of RSV through hydrolysis by esterase enzymes of **1**, was confirmed using digestion experiments with cell lysates. Moreover, the effect of **1** on tyrosin activity, cellular viability, and cellular melanin synthesis was investigated. In particular, compound **1** was able to affect the melanin synthesis in a reconstituted human skin model, confirming its potential as an anti-melanogenic agent for cosmetic use [161].

Later, the same research group evaluated the skin-whitening effect of RSV triacetate in human (**1**) [156]. Initially, the primary skin irritation test revealed that **1** did not induce any skin irritation at the tested concentration, while RSV caused a weak response at the same dose. Subsequently, the efficacy in skin-whitening of a topical formulation containing 0.4% of **1** was investigated using two different human models. Compound **1** showed a statistically significant skin-whitening effect between the test group and the control group, in both the artificial tanning model and the hyperpigmentation model. Due to the better stability and the less toxicity of the topical formulation containing **1**, if compared with a similar formulation containing the same concentration of RSV, the authors suggested the use of **1** as a cosmetic ingredient in skin-whitening products, even though RSV is more effective in inhibiting melanin synthesis [156,161].

With a similar purpose, the triglycolate derivative of RSV (**41**, Figure 15) was developed [162]. In this study, the anti-melanogenic effect of the new RSV prodrug was compared to that of RSV, glycolic acid, arbutin, and the previously reported RSV triacetate (**1**). The inhibition of the in vitro catalytic activity of human Tyrosinase (TYR) of RSV, **1**, and **41** was higher than glycolic acid and arbutin, respectively. Similarly, the anti-melanogenic effect of **41** was comparable to that of RSV and **1**, as well as the suppression activity of melanogenic enzymes and the inhibition of catalytic activity of TYR enzyme. Because the solubility of **41** in many cosmetic solvents was higher than **1**, and due to the more suitable chemical structure of **41** in terms of hydrophobic/hydrophilic balances, the authors speculated about the advantages of using **41** instead of **1** in topical formulations [162].

Very recently, the skin-depigmenting efficacy of **41** in vitro has also been confirmed in vivo [159]. Specifically, the visual assessment of pigmentation degree, instrumental analysis of melanin index, skin lightness, and skin color after eight weeks of treatment with a topical formulation containing 0.4% of **41** were assessed in humans. The results suggested depigmentation enhancement of the skin in the test group compared with the control group, with no adverse skin reactions. The stability of **41** against high temperature and sunlight was intermediate between RSV and **1** [159]. Although these data showed a skin-whitening effect of **41**, a direct comparison of the efficacy and potency of **41** vs. **1** was not reported.

Lephart et al. described a regioselective one-step esterification of RSV, which led to the synthesis of 4′-acetoxy RSV derivative (**34**, Figure 14) [157]. The effect of **34** in regulating skin gene expression, was also investigated using several in vitro human dermal models, in particular, the following effects were evaluated: (1) stimulation of gene expression of SIRT1 and ECM proteins; (2) expression of skin antioxidant and skin aging biomarkers; (3) influence on the expression of skin inflammatory biomarkers. Briefly, compound **34** showed ability in increasing gene expression of the anti-aging factor (SIRT1), the extracellular matrix proteins, and the antioxidants biomarkers. On the other hand, **34** was able to significantly down-regulated gene expression of inflammatory and skin-aging molecules, including IL-1, IL-6, IL-8, COX-2, and TNFRSF (the tumor necrosis factor receptor superfamily) [157]. Moreover, comparison with the previously reported data of RSV effects [163], on the same battery of assays, revealed that **34** was about 1.6-fold more potent than RSV, suggesting a marginal increase in effectiveness [157]. Finally, the authors hypothesized that good stability and a low metabolism occurred for **34** in human skin, proposing compound **34** as an RSV analog rather than a prodrug. Nevertheless, they did not perform stability and bioavailability studies neither in vitro nor in vivo; therefore these conclusions were not confirmed.

In another study, the same research group tested the in vitro activity on human skin genes expression of five different RSV derivatives (**37**, **38**, **42**–**44**, Figure 16) and RSV, which was used as a standard [158]. The structural modifications regarded the introduction of different acyl groups (i.e., isobutyryl, butyryl, palmitoyl, and acetyl) in different positions of the RSV core (4′-position, 3-position, and 3,5-positions). Among them, isobutyrate and butyrate derivatives (**37** and **38**, respectively) showed a significant effect on stimulation of genes expression (SIRT1, collagens, and extracellular matrix proteins), and significant inhibition of inflammatory and aging biomarkers. More in detail, examining the series, when compared to the RVS effects the gene expression level was ranked from highest-to-lowest as follows: **37** > **38** > **44** > **43** > **42**. Overall, these data proved the potential utility, in vitro, of **44** and **45** in topical formulations (e.g., anti-aging products) [158]. This study was mainly focused on the comparison of potency and biological effects of RSV analogs versus RSV and, unfortunately, the lack of metabolic stability studies did not allow a direct correlation between physico-chemical properties and biological activity; thus, further investigations are needed for translating into actual applications in vivo.

Finally, other RSV derivatives have been prepared and patented as ingredients for cosmetic preparations [164,165,166,167,168].

## 4. Conclusions

To summarize, several functionalizations on the phenolic hydroxyls of RSV have been reported so far, and a variety of chemically different capping groups and promoieties have been used to modulate absorption and release of the parent compound, as well as to improve its biological activity.

Despite some of the RSV prodrugs indicating significant chemical stability, most of them showed poor pharmacokinetic properties, which could drastically reduce their practical use in vivo. Several chemical and biological aspects are involved in prodrug development, thus balancing the ratio between optimal chemical stability and proper absorption is quite challenging. The in vitro studies are the majority of the characterization performed on RSV prodrugs, which represent the starting point for testing the proof of concept. In most of the case, these preliminary studies are essential to determine if it is worth to perform more in-depth studies; however, this step should not limit the full characterization of a new chemical entity. Unfortunately, due to the high level of expertise and costs required for developing prodrugs, a limited number of in vivo evaluations have been addressed so far. This lack of in vivo studies is undoubtedly the most limiting factor for the translational into clinical. In contrast, several RSV loaded delivery systems have been tested in suitable animal models, providing encouraging outcomes that point out their ability to improve RSV effectiveness by different administration routes. Therefore, integration of both the prodrug approach and the formulation technologies might represent a valid solution to solve the bioavailability issue of RSV, by exerting a complementary role that is useful for overcoming the specific limitations of each method.

## Figures and Tables

**Figure 1 antioxidants-08-00244-f001:**
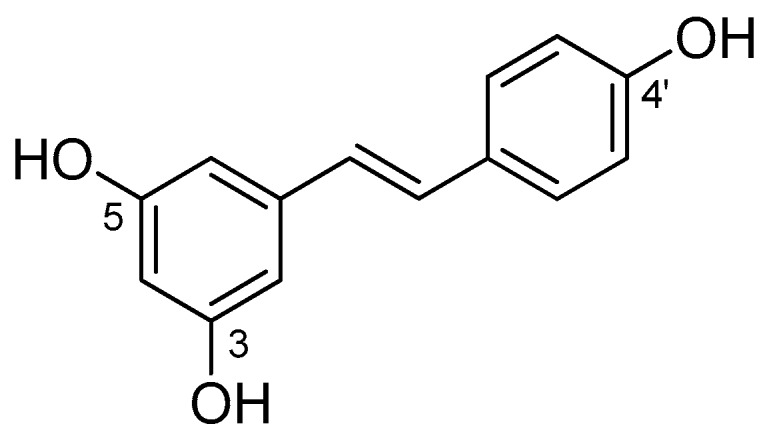
Chemical structure of *trans*-RSV (3,5,4′-trihydroxy-*trans*-stilbene).

**Figure 2 antioxidants-08-00244-f002:**
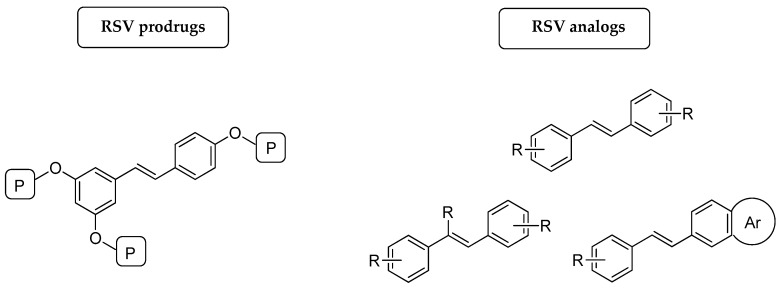
General structures of RSV prodrugs and RSV analogs.

**Figure 3 antioxidants-08-00244-f003:**
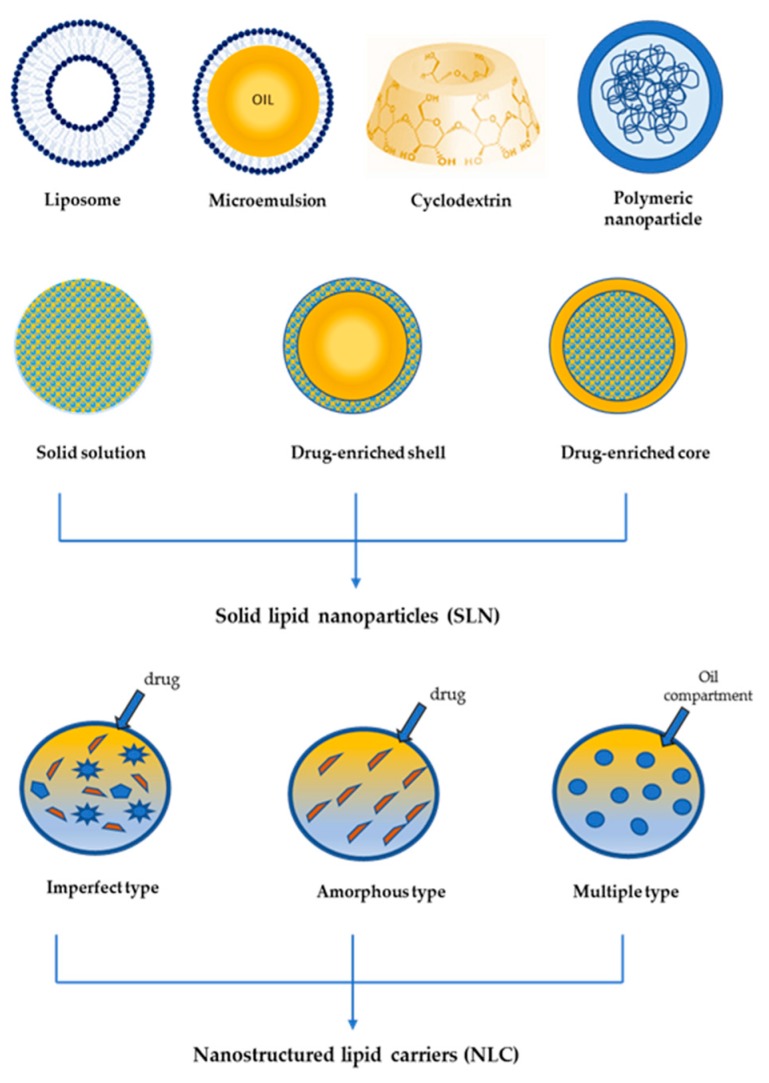
Schematic representation of drug delivery systems.

**Figure 4 antioxidants-08-00244-f004:**
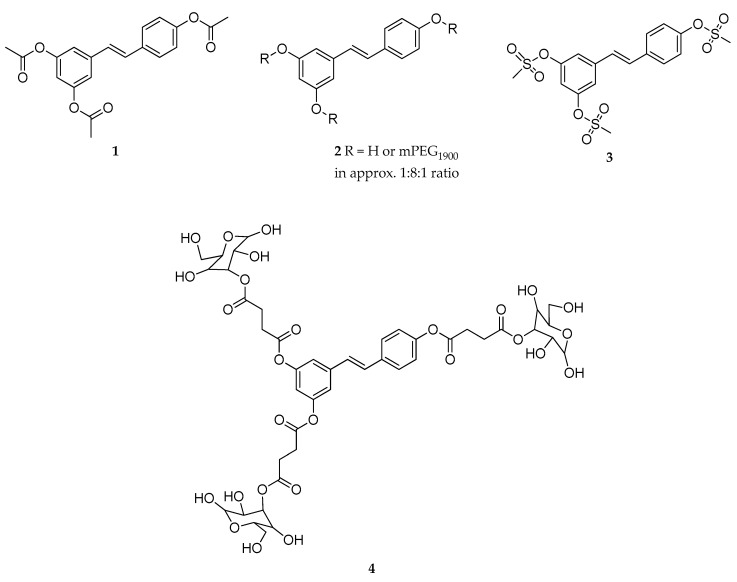
Chemical structures of RSV triacetate (**1**), RSV tri-mPEG_1900_ (**2**), RVS trimesylate (**3**), and 3,5,4′-tri(α-d-glucose-3-*O*-succinyl)-resveratrol (**4**).

**Figure 5 antioxidants-08-00244-f005:**
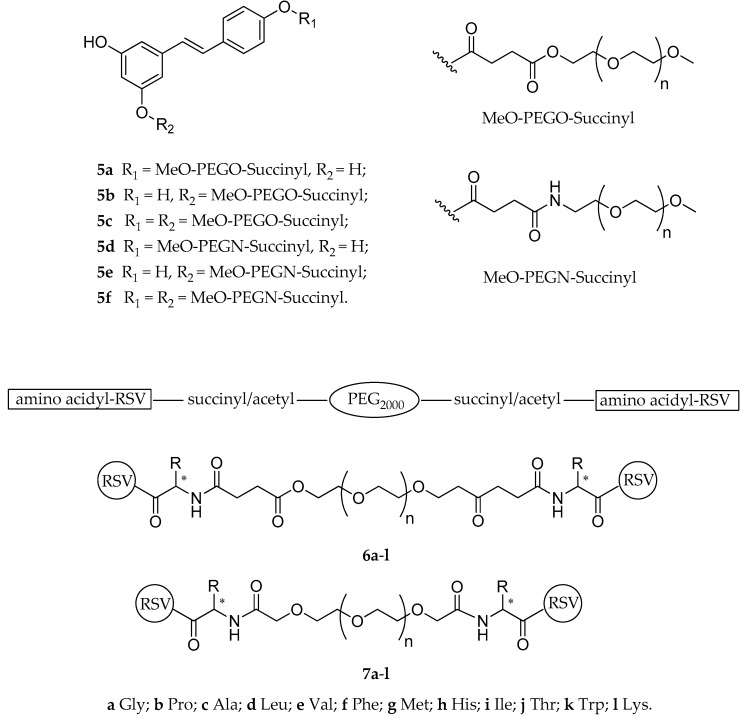
Chemical structures of PEG-resveratrol prodrugs **5a–f**, **6a–l**, and **7a–l**. * = chiral center

**Figure 6 antioxidants-08-00244-f006:**
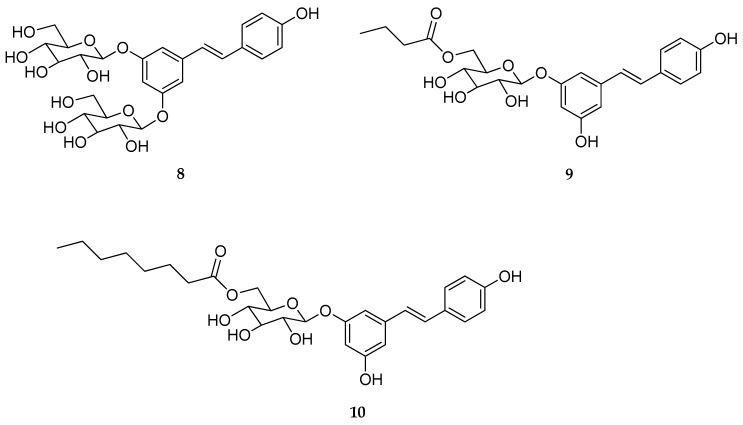
Chemical structures of RSV-3,5-di-*O*-β-d-glucopyranoside (**8**), RSV-3-*O*-(6′-*O*-butanoyl)-*β*-d-glucopyranoside (**9**), RSV-3-*O*-(6′-*O*-octanoyl)-*β*-d-glucopyranoside (**10**).

**Figure 7 antioxidants-08-00244-f007:**
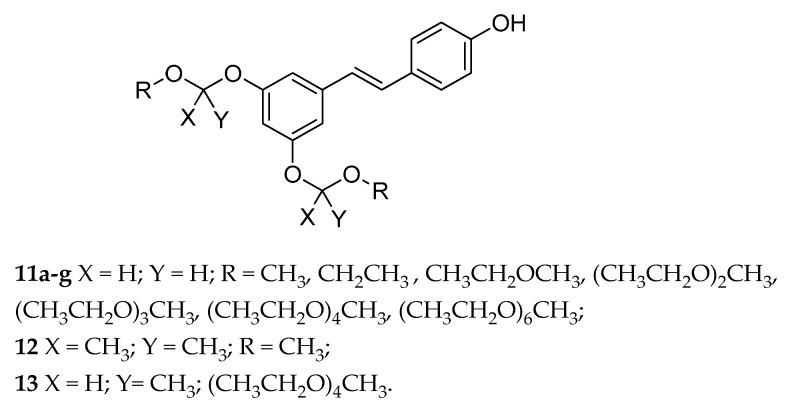
Chemical structures of acetal derivatives of RSV (**11**–**13**).

**Figure 8 antioxidants-08-00244-f008:**
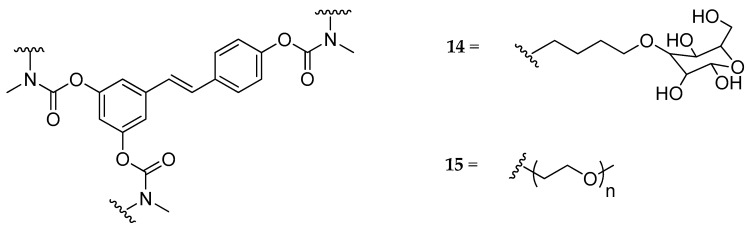
Chemical structures of *N*,*N*-disubstituted carbamoyl-RSV derivatives **14**, and **15**.

**Figure 9 antioxidants-08-00244-f009:**
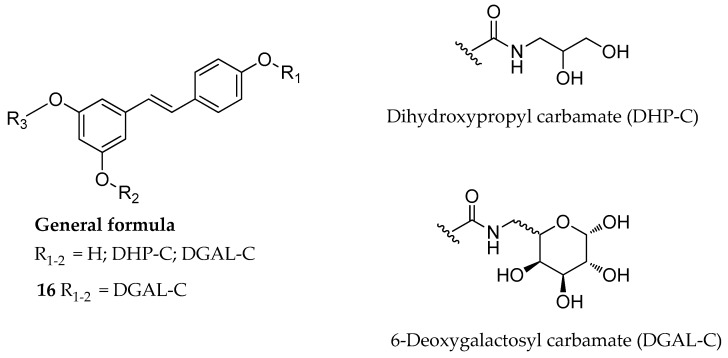
General formula of *N*-monosubstituted carbamoyl-RSV derivatives, and chemical structure of **16**.

**Figure 10 antioxidants-08-00244-f010:**
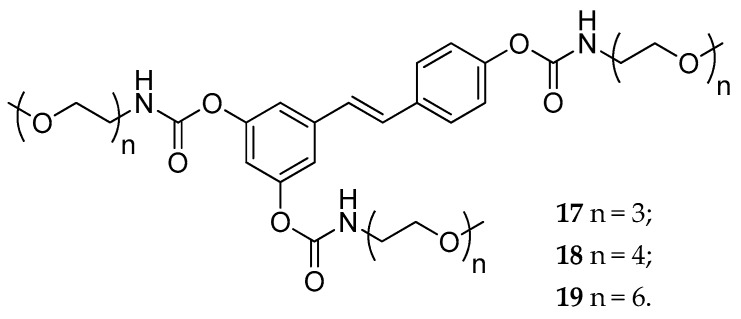
Chemical structures of methoxy-oligo(ethylene glycol)-carbamate substituted prodrugs, **17**–**19**.

**Figure 11 antioxidants-08-00244-f011:**
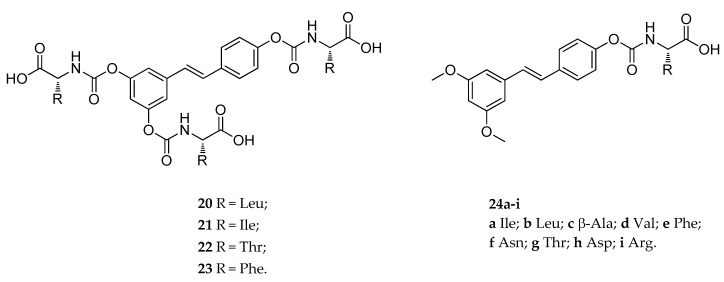
Chemical structures of *N*-monosubstituted carbamate amino acidic prodrugs **20**–**23**, and **24a**–**i**.

**Figure 12 antioxidants-08-00244-f012:**
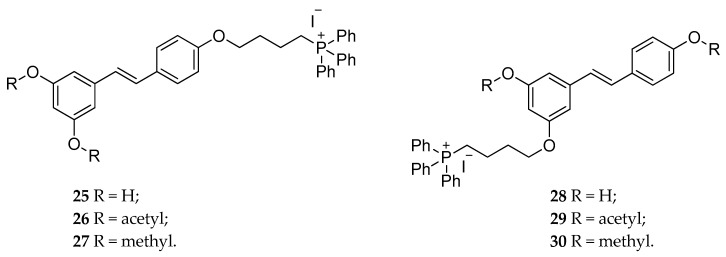
Chemical structures of RSV-triphenylphosphonium derivatives **25**–**30**.

**Figure 13 antioxidants-08-00244-f013:**
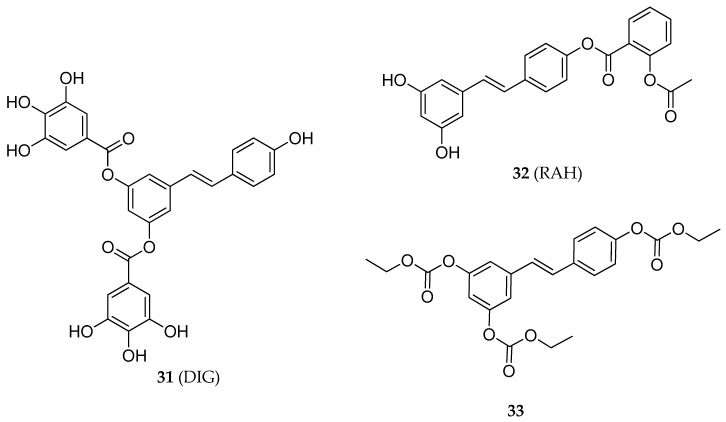
Chemical structures of 3,5-*O*-digalloylresveratrol (DIG) **31**, RSV-aspirin hybrid (RAH) **32**, and compound **33**.

**Figure 14 antioxidants-08-00244-f014:**
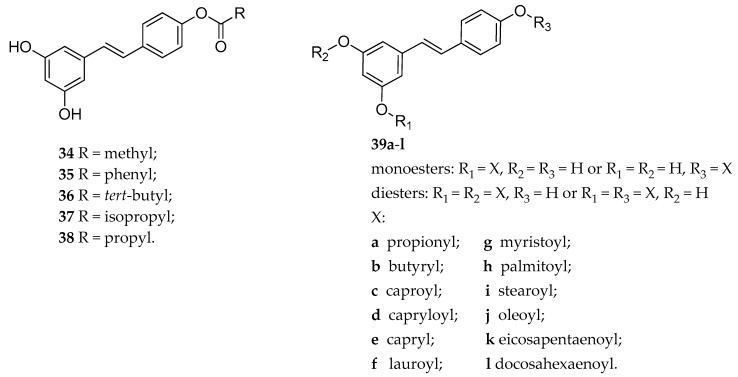
Chemical structures of 4′-RSV ester derivatives **34**–**38**, and mono and diester RSV derivatives **39a–l**.

**Figure 15 antioxidants-08-00244-f015:**
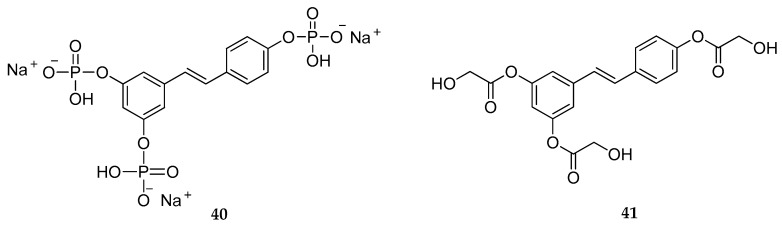
Chemical structures of RSV triphosphate (**40**) and RSV triglycolate (**41**).

**Figure 16 antioxidants-08-00244-f016:**
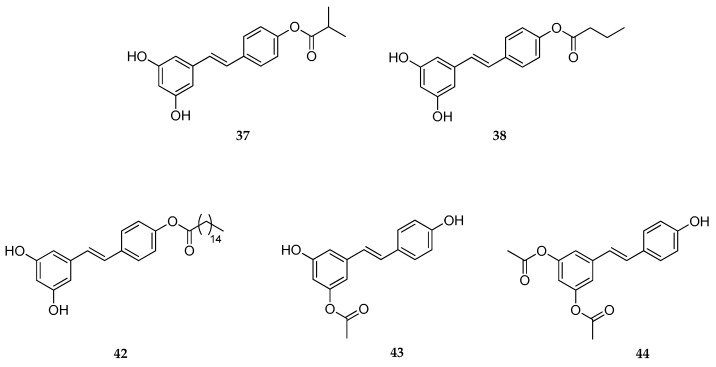
Chemical structures of RSV 4′-isobutyrate (**37**), RSV 4′-butyrate (**38**), RSV 4′-palmitoate (**42**), RSV 3-acetate (**43**), and RSV 3,5-diacetate (**44**).

**Table 1 antioxidants-08-00244-t001:** Physicochemical properties of *trans*-RSV.

Molecular Descriptors ^1^	
Molecular weight (MW)	228.25
Calculated LogP (cLogP)	3.40
Hydrogen Bond Donors (HBD)	3
Hydrogen Bond Acceptors (HBA)	3
Rotatable Bonds Number (RBN)	2
Topological Polar Surface Area (TPSA)	60.69

^1^ Calculated using Marvin 17.21.0, ChemAxon (ChemAxon Ltd., Budapest, Hungary) (https://www.chemaxon.com).

**Table 2 antioxidants-08-00244-t002:** Oral resveratrol (RSV) drug delivery systems. NP = nanoparticles; PLGA = dl-lactide-*co*-glycolide; SLN = solid lipid nanoparticles.

Nanocarriers	Study	Animal Model	Outcomes	Ref.
PLGA NP	RSV pharmacokinetics, in vivo biodistribution, single-pass intestinal perfusion	rats	↑ RSV oral bioavailability in comparison to the free drug	[85]
Galactosylated PLGA NP (GNPs)	oral bioavailability	rats	↑ RSV oral bioavailability, ↑ intestinal permeability and transcellular transport of RSV, ↑ anti-inflammatory activity in RAW 264.7 cells model	[86]
NP based on chitosan derivatives	antioxidant activity and in vivo bioavailability	rats	↑ RSV water solubility, ↑ antioxidant activity, ↑ bioavailability	[87]
SLN *N*-trimethyl chitosan conjugated with palmitic acid	oral bioavailability	Balb/c mice	↑ RSV bioavailability, ↑ ability to prevent RSV enzymatic and/or chemical degradation	[88]
SLN coated with poloxamer 188	oral bioavailability	in rats	↑ RSV effectiveness after oral dosing, ↑ RSV bioavailability	[89]
SLN and NLC	oral bioavailability	in vitro study	RSV controlled release Prevention of RSV degradation	[90]
Layer by layer (LbL) NP	pharmacokinetic study	Wistar rats	↑ RSV bioavailability and chemical stability	[91]

**Table 3 antioxidants-08-00244-t003:** Parental resveratrol (RSV) drug delivery systems. DTX = docetaxel; LPNs = lipid-polymer hybrid NP; NP = nanoparticles; PEG-PLA = polyethylene glycol - polylactic acid; i.v.= intravenous; i.p. intraperitoneal; BBB = blood brain barrier; PM = polymeric micelles; PCT = paclitaxel; (pHO-1) = heme oxygenase-1 gene.

Nanocarriers	Study	Animal Model	Outcomes	Ref.
RSV and DTX co-encapsulated into LPNs	treatment of lung cancer	i.v. injection in mice in vitro (cells HCC827 and NCIH2135)	↓ tumor growth and size, ↓ the viability of tumor cells	[93]
PEG-PLA NP	cancer treatment	in vitro assays on CT26 colon cancer cell i.v. administration in tumor-bearing mice	↓ reduction of cell number and colony forming, ↓ tumor growth, ↑ survival time of mice, ↑ RSV stability	[94]
PEG-PLA NP including transferrin (Tf)	treatment of glioma	i.p. administration in C6 glioma-bearing rat models	↑ anti-cancer activity, ↓ tumor volume with a concomitant increase of survival time	[95]
lipid core nanocapsules	treatment of glioma	i.p. administration in rats bearing brain-implanted C6 gliomas	↓ decrease of tumor size, ↑ RSV transportation across the BBB, ↓ RSV binding to plasma protein	[96]
PM RSV and DTX co-loaded	in vitro cytotoxicity pharmacokinetic	MCF-7 cells, i.v. administration in rats	↑ AUC values	[98]
PM RSV and PTX co-loaded	antitumor activity	PTX-resistant human lung adenocarcinoma epithelial (A549/T) cell line and mice sarcoma 180 (S180) cells, i.v. injection in S180 solid tumor bearing mice	↑ inhibition of tumor growth	[99]
PM RSV co-loaded with (pHO-1)	treatment of acute lung injury.	Inhalation in Balb/c mice	inhibits the nuclear translocation of NF-kB, ↓ pro-inflammatory cytokines in lungs	[100]
Gelatin NP	bioavailability anti-proliferative effect in NCI-H460 lung cancer cells	i.v. injection in mice	↑ bioavailability, ↑ anti-proliferative effect	[101]
chitosan-coated lipid NP	brain delivery	inhalation in rat	↑ RSV concentration in cerebrospinal fluid	[102]
cyclodextrins	Bioavailability	i.v. and oral administration in rats	No modifications of bioavailability	[103]

**Table 4 antioxidants-08-00244-t004:** Topical resveratrol (RSV) drug delivery systems. SLN = solid lipid nanoparticles; NLC = nanostructured lipid carriers; UDL = ultra-deformable liposomes; 5-FU = 5-flurouracil.

Nanocarriers	Drug	Study	Outcomes	Ref.
SLN	RSV	in vitro penetration (pig skin)	↑ RSV photostability, ↑ accumulation in the skin, ↑ anti-lipoperoxidative activity	[104]
NLC	RSV	in vitro permeation (human skin)	↑ RSV skin permeation, ↑ RSV topical effectiveness	[105]
SLN	RSV	in vitro permeation (human skin) in vivo studies (ICD-induced BALB/c mice)	↑ RSV increase of its retention in the skin layers (epidermis and dermis) ↓ tissue edema	[106]
SLN	RSV	skin permeation (pig skin) tyrosinase activity	↑ percentage of tyrosinase inhibitory activity,↑ skin permeation	[107]
SLN/NLC	RSV	skin hydration healthy volunteers	SLN ↑ skin hydration in comparison to NLC	[108]
SLN/NLC	RSV	in vitro penetration studies (rat skin)	SLN ↑ RSV accumulation in the epidermis, NLC ↑ amount of RSV in the dermis	[109]
Chitosan- coated Liposomes	RSV	in vitro permeation (mouse skin)	↑ skin permeation	[110]
Chitosan- coated Liposomes	RSV	in vitro permeation (mouse skin)	↓ vaginal inflammation and infections, ↑ antioxidant and anti-inflammatory activities	[111,112]
UDL	Psoralen and RSV	antioxidant assays B16F10 cell line	↑ tyrosinase activity, ↑antioxidant activity	[113]
UDL	RSV/5-FU	in vitro permeation (human skin) anti-cancer activity on SK-MEL-28 and Colo-38 cells.	Drug accumulation in the deeper skin layers ↑ anti-cancer activity	[114]

**Table 5 antioxidants-08-00244-t005:** Summary of RSV prodrugs for systemic delivery and their stability properties.

Type of Prodrug	Promoiety	Linker	Chemical Stability	Stability in Blood	Solubility in Water	Ref.
Triester (**1**)	acetyl	-	slowly hydrolyzed	rapidly hydrolyzed	poorly soluble	[121,122]
Tri-mPEG (**2**)	mPEG ^a^	-	slowly hydrolyzed	rapidly hydrolyzed	poorly soluble	[121]
Trimesylate (**3**)	mesyl	-	not hydrolyzed	not hydrolyzed	extremely soluble	[121]
Triester (**4**)	α-d-glucose	succinyl	mostly stable	rapidly hydrolyzed	highly soluble	[123]
Diesters (**5a–f**)	mPEG ^a^	succinylester or succinylamide	not reported	not reported	not reported	[124]
Esters (**6a–l**, **7a–l**)	PEG-succinyl or PEG-acetyl	various amino acids	not reported	hydrolyzed	highly soluble	[125]
Mono-*O*- glucoside and di-*O*-glucoside (**8**–**10**)	*β*-d-glucose	-	stable	stable	not reported	[126]
Acetals (**11a–g**, **12**, **13**)	OEG ^b^	-	variable	variable	from poor to highly soluble	[128]
Tri-*N*,*N*-di- substituted carbamate, (**14**, **15**)	butyl-glucosyl or mPEG ^a^	-	-	not hydrolyzed	highly soluble	[129]
Mono-, di- or tri-*N*-mono- substituted carbamate, (e.g., **16**)	glycerol or galactose	-	slowly hydrolyzed	slowly hydrolyzed	highly soluble	[130]
tri-*N*-mono- substituted carbamate, (**17**–**19**)	OEG ^b^		slowly hydrolyzed	fast hydrolyzed	not reported	[131]
tri-*N*-mono- substituted carbamate (**20**–**23**)	various amino acids	carbamoyl	slowly hydrolyzed	suitable hydrolyzed	not reported	[133]
mono-*N*-mono- substituted carbamate (**24a–i**)	various amino acids and metyl	carbamoyl	slowly hydrolyzed	suitable hydrolyzed	not reported	[133]

^a^ Methoxypolyethylene glycol; ^b^ oligoethyleneglycol.

**Table 6 antioxidants-08-00244-t006:** Summary of RSV prodrugs for systemic delivery and their biological effects.

Type of Prodrug	Compounds	Effects	Ref.
RSV-triphenylphosphonium	**25**–**30**	cytotoxicity effect in C-26 murine colon cancer cell line	[134]
3,5-RSV diester	**31**	apoptotic effect in HT-29 human colon cancer cells line	[141]
4′-RSV ester	**32**	growth inhibition in HCT-116 and HT-29 human colon cancer cells line	[145]
RSV tricarbonate	**33**	antiproliferative and pro-apoptotic activities in Jurkat T-cells.	[148]
4′-RSV ester	**34**–**36**	anti-depressant activity of **33** in rats (Porsalt forced-swim test)	[149]
4′-RSV ester	**36**–**38**	decreasing cell viability in MDA-MB-231 cancer cells line	[150]
RSV mono and diester	**39a–l**	antioxidant activity in both DPPH and ABTS radical cation scavenging assays	[151]
RSV mono and diester	**39a–l**	inhibition of hydroxyl radical-induced DNA scission	[152]

**Table 7 antioxidants-08-00244-t007:** Summary of RSV prodrugs for dermal delivery and their cosmetic application.

Type of Prodrug	Compounds	Cosmetic Application	Ref.
Triphosphate	**40**	not stated	[160]
Triacetate	**1**	anti-melanogenic agents/whitening effect	[156,161]
Triglycolate	**42**	anti-melanogenic agents/whitening effect	[159,162]
4′-Acetate	**43**	skin antioxidant, skin anti-aging	[157]
4′-, 3-, 3,5- Esters	**37, 38, 42**–**44**	skin antioxidant, skin anti-aging	[158]

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
