# Peer review of "Strategies to Improve Resveratrol Systemic and Topical Bioavailability: An Update"

_antioxidants, 2019, doi:10.3390/antiox8080244_

Reviewer 1 Report

Dear Editor,

The current manuscript described a review on strategies to improve resveratrol oral bioavailability. Review is very interesting, and it will be helpful for ongoing resveratrol research. The subject of considered manuscript fitting well with the scope of "Antioxidant" and indicated excellent scientific value.

My comments are as follows:

1.    Manuscript has several grammatical errors. It should be checked by a native English speaker.

2.    Recalculate/recheck number of hydrogen bond donors of trans-RSV in Table 1.

3.    Include pharmacokinetic parameters of resveratrol only. It will be helpful in its comparison with formulation and prodrug approach.

4.    Is there any successful approach (formulation or prodrug) which has been translated into clinical setting? Please highlight these clinically successful approaches in conclusion section.

Regards

Author Response

1.    Manuscript has several grammatical errors. It should be checked by a native English speaker.

Answer

As requested, a native English speaker checked the manuscript.

2.    Recalculate/recheck number of hydrogen bond donors of trans-RSV in Table 1.

Answer

We thank the reviewer for the comment. We rechecked all values reported in Table 1 and we corrected the number of hydrogen bond donors.

3.    Include pharmacokinetic parameters of resveratrol only. It will be helpful in its comparison with formulation and prodrug approach.

Answer

As requested, we inserted the pharmacokinetic parameters of resveratrol. Therefore, at line 70 (revised version) we added the following sentences:  “As reported in the literature, after oral administration, RSV is quickly absorbed at gastrointestinal level but its bioavailability is lower than 1% due to an extensive first pass effect [53]. Walle et al. reported that RSV plasma half-life was 9.2 hours and they observed a first peak of RSV plasma concentration 1 hour after an oral dose of 25 mg and a second peak after 6 hours [55].

We amended the reference list accordingly.

4.    Is there any successful approach (formulation or prodrug) which has been translated into clinical setting? Please highlight these clinically successful approaches in conclusion section.

Answer

We thank the reviewer for the interesting comment but unfortunately, to the best of our knowledge, so far no clinical trial has been set on the prodrugs and drug delivery systems illustrated in this manuscript.

Reviewer 2 Report

Overall an excellent review in the field of RSV and its derivatives due to the biological effects. The authors also pay attention to improve RSV effectiveness, as well as the chemical modifications. I believe the readers of antioxidants will be interested in this content and recommend the acceptance of this manuscript.

Author Response

Overall an excellent review in the field of RSV and its derivatives due to the biological effects. The authors also pay attention to improve RSV effectiveness, as well as the chemical modifications. I believe the readers of antioxidants will be interested in this content and recommend the acceptance of this manuscript.

Answer

We would like to thank the reviewer for reviewing our manuscript. We have highly appreciated the reviewer’s comments.